# Unmasking individual differences in adult reading procedures by disrupting holistic orthographic perception

**Elizabeth A. Hirshorn** [1] *, **Travis Simcox** [2,3,4], **Corrine Durisko** [2], **Charles A. Perfetti** [2,3,4], **Julie A. Fiez** [2,3,4]

**1** Department of Psychology, SUNY New Paltz, New Paltz, New York, United States of America, **2** Learning Research and Development Center, University of Pittsburgh, Pittsburgh, PA, United States of America, **3** Department of Psychology, University of Pittsburgh, Pittsburgh, PA, United States of America, **4** The Center for the Neural Basis of Cognition, Pittsburgh, PA, United States of America

* hirshore@newpaltz.edu

**Data Availability Statement:** All relevant data are within the manuscript and its Supporting Information files.

## Abstract

Word identification is undeniably important for skilled reading and ultimately reading comprehension. Interestingly, both lexical and sublexical procedures can support word identification. Recent cross-linguistic comparisons have demonstrated that there are biases in orthographic coding (e.g., holistic vs. analytic) linked with differences in writing systems, such that holistic orthographic coding is correlated with lexical-level reading procedures and vice versa. The current study uses a measure of holistic visual processing used in the face processing literature, orientation sensitivity, to test individual differences in word identification within a native English population. Results revealed that greater orientation sensitivity (i.e., greater holistic processing) was associated with a reading profile that relies less on sublexical phonological measures and more on lexical-level characteristics within the skilled English readers. Parallels to Chinese procedures of reading and a proposed alternative route to skilled reading are discussed.

## Introduction

Visual word identification serves as the foundation for skilled reading comprehension. Word identification occurs when the phonological and semantic representations of a word are accessed in response to seeing a visual word form. In an alphabet like English, there is evidence that visual words are coded both holistically and analytically, with holistic representations capturing the mapping between a specific visual form and its lexical equivalent, and analytic coding capturing the mapping between sublexical orthographic units (e.g., graphemes, bigrams, etc.) and their phonological equivalents. The current study uses a simple behavioral marker of sensitivity to the spatial orientation of print to investigate individual differences in orthographic coding amongst native English speakers, and their relationship to the reading procedures that support word identification and comprehension.

**Funding:** Eunice Kennedy Shriver National Institute of Child Health and Human Development under Award Number R01HD060388 (to JAF) supported this work. The funder had no role in the study design, data collection and analysis, decision to publish, or preparation of the manuscript.

**Competing interests:** The authors have declared that no competing interests exist.

## Holistic versus analytic orthographic coding

Orientation manipulations have historically been used to differentially disrupt visual object recognition of items that are processed more holistically (e.g., faces) compared to items that are processed more analytically, or in a piecemeal manner. An atypical orientation is thought to disproportionately affect holistically processed objects. In some accounts, this is because the holistic process cannot be applied when the object is presented unconventionally [1, 2]. This forces a switch in recognition strategy towards a more analytical or feature-based approach, which is less than optimal and more error prone for holistically processed objects like faces [3]. Others have been argued that inverted faces are eventually processed holistically [4], but even in this case one would still expect that a 'holistic' strategy would take longer and potentially be more error prone, because the presented orientation is suboptimal for holistic processing.

Similarly, individual biases for holistic versus analytic orthographic coding can be determined by manipulating the orientation of visual word forms and measuring the impact it has on word recognition. The logic rests upon the same concepts that apply in the face processing literature [5]. For example, the idea that an atypical orientation forces a more analytical approach in word identification is supported by the emergence of a word length effect for rotated words [6, 7], but not upright words. The reasoning is that if a typical left to right decoding is disrupted due to atypical orientation of a word (Coltheart et al., 2001), an analytical approach becomes necessary (i.e., relying more on sub-word units) and longer words should take more time to decode than shorter words [8]. This approach of disrupting the typical presentation has recently been used to measure holistic coding in visual word processing [9, 10] and uncover cross-linguistic biases in orthographic coding [11–13].

## Implications of orthographic coding procedures for word identification

The question of interest for the current study is whether individuals with potential systematic differing preferences for holistic versus analytic orthographic coding exhibit corresponding differences in the procedures and skills they use for word identification. Evidence for a link between orthographic coding and the procedures used for word identification comes from cross-linguistic studies comparing Chinese and Korean bilinguals reading English. Chinese is a morpho-syllabic writing system, and so sublexical orthographic coding and mapping to phonology is arguably less useful for word identification, as compared to the holistic coding and mapping of characters to their morphemic forms, although the extent of reliance on holistic processing of Chinese characters might depend on an individual's writing experience [14]. Korean, in contrast, is a highly consistent alphabetic writing system in which reading instruction emphasizes the decoding of printed words based on sublexical orthographic-phonological correspondences. Three studies have used visual form manipulations to investigate whether Chinese and Korean indviduals bring biases from their native writing system to English reading [11–13]. In all three of these studies, Chinese-English bilinguals exhibited greater holistic orthographic coding and a bias towards lexical-level processing to support word identification, while Korean-English bilinguals exhibited greater analytic orthographic coding and a bias towards sublexical and phonological processing to support word identification. For example, Ben-Yehudah and colleagues (2018) found that Chinese-English bilinguals' naming times were more sensitive to word inversion and lexical frequency, whereas Korean-English bilinguals were relatively unaffected by word inversion and more sensitive to spelling-to-sound consistency [8]. Thus, prior cross-linguistic research provides evidence that individuals can exhibit differing biases for holistic versus analytic orthographic coding of English, and this differing bias is associated with a preference for lexical versus sublexical reading procedures.

The current study extends this prior cross-linguistic work by testing for similar patterns of individual differences amongst native English speakers. The study uses an orientation manipulation as a functional marker defining two groups who show a bias toward either holistic or analytic orthographic coding. Then, to test for associated biases toward lexical vs. sublexical reading procedures, the two groups are compared for their sensitivity to psycholinguistic properties of words in an overt naming task. Specifically, the design of the naming task manipulates the following word properties: 1) lexical frequency (how often a word appears in databases of written text), 2) imageability, 3) consistency/regularity (whether the pronunciation of a word is predictable based upon the spelling of its rime body (consistency); whether it follows the grapheme-phoneme correspondence rules of the writing system (regularity)), 4) length, 5) bigram frequency, and 6) biphone frequency [15].

The predicted outcome is that individuals with a bias toward holistic orthographic coding (i.e., high orientation sensitivity) will have a bias toward lexical reading procedures, as indicated by a: 1) heightened sensitivity to frequency, because frequency effects are widely regarded as a measure of lexical-level influences [8, 16–18]; 2) greater sensitivity to imageability, because it is a semantic measure that is inherently processed at a larger grain size [e.g., morpheme or whole word; 18, 19], and 3) reduced sensitivity to consistency/regularity effects in addition to bigram and biphone frequency, because they are widely regarded as measures of sublexical influences on orthographic-phonological mapping [8, 16, 20]. An increased length effect is expected for atypically presented words [6, 7] for individuals with a bias toward a lexical-level reading procedure, as an atypical presentation should necessitate a greater reliance on sublexical processing (i.e., their less-preferred reading procedure). Opposite reading patterns would be expected for individuals with a bias towards sublexical reading approaches.

## Implications of orthographic coding procedures for lexical representation

Another component that influences reading procedures is the structure of one's lexical representations, or lexical integration. Lexical representations consist of three constituents: orthography, phonology, and semantics and individual differences in the quality of these knowledge components affect reading processes [21]. The structure of lexical representations can be quantitatively described using a factor analysis to capture the correlational structure of performance on tasks that emphasize orthographic, phonological, and semantic knowledge. Phonological decoding, or the correspondence between orthography and phonology that allows a reader to correctly pronounce a word [22–24], has been highlighted as a particularly important foundational skill in word identification. Phonological decoding requires knowledge of sublexical orthographic-to-phonological regularities to pronounce words, and is commonly measured using a nonword reading task. This line of reasoning leads us to test whether individuals with more holistic vs. analytic orthographic coding exhibit structural differences in their lexical representations. We hypothesize that individuals with greater holistic orthographic coding (and a bias towards lexical reading procedures) will have a lexical representational structure without tight correlations to measures of sublexical phonological decoding.

## Summary of study

In summary, the current study extends what we know about the link between orthographic coding and reading procedures from cross-linguistic work to individual differences within native English readers. The overarching hypothesis is that native English readers who show more sensitivity to atypical orientations of printed word forms are more reliant on holistic orthographic coding, and in turn possess a distinctive reading profile. We expect this reading profile to be similar to Chinese-English readers, showing a greater reliance on lexical-level

reading procedures and reduced reliance on phonological decoding in the representational structure of word-level processing.

## Materials and methods

### Group definition criterion

Participants were recruited from a database of 411 individuals interested in study opportunities. Participants completed a series of screening tasks on a computer. The screening was limited to tasks that participants could respond to with button presses. Non-monolingual individuals and participants who identified as having trouble reading or a history of reading disorder were not eligible.

Participants were initially identified using data from a lexical decision task (see below), which included words with typical and atypical (180° rotation, see Fig 1) orientation. To focus on individuals with average reading ability, participants outside the 25th to 75th percentile in their median reaction time (RT) to typically presented words were removed, leaving 203 from an initial 411 potential participants. Accuracy and reaction times were calculated. Incorrect trials were removed from the reaction time analyses. The orientation sensitivity of each participant was calculated as the ratio of median RT for inverted stimuli divided by the median RT for upright stimuli. The median ratio of the remaining participants was 1.42 and the standard error (SE) was .037. Group cutoffs were delineated as the median ± 2 SE (rounded to 1.5 and 1.35). Thus, participants with higher sensitivity to atypical orientation (HS) were defined as those individuals with RTs for atypically oriented stimuli (words and nonwords) that were at least 1.5 times greater than typical stimuli. Participants with lower sensitivity to atypical orientation (LS) were defined as those individuals with RTs for atypically oriented stimuli that were less than 1.35 times that of typical stimuli. Participant attrition, due to graduation since participation in initial screening and eligibility requirements for a parallel imaging study (beyond the scope of the current study), also reduced the potential participant pool. A total of 31 eligible participants were run in a second behavioral testing session. A subset of these participants (N = 22) was run in a companion imaging study [25].

In order to ensure robust group assignment, sensitivity to orientation was also computed in an overt word naming task (see Materials, Overt word naming) that manipulated atypical orientation. This task was completed in the second behavioral session. Six participants whose overt naming scores were neither above nor below the median orientation sensitivity that was consistent with their initial group assignment based on the lexical decision task were removed.

### Participants

All final participants were native monolingual English speaking undergraduates with no reported history of hearing or vision issues, learning or reading difficulties, drug or alcohol abuse, mental illness, or neurological problems. All final participants scored above the 20th percentile of Raven's Matrices [26] and were dominantly right-handed. All participants provided informed consent and were given class credit for their first session and were monetarily compensated for an additional second session. Fourteen Lower Sensitivity Readers (LS) (2 males; mean age = 20.1 years, SD = 2.62) and 11 Higher Sensitivity Readers (HS) (4 males; mean age = 19.3 years, SD = 0.65) participated in the final experiment (see Table 1). All participants provided informed consent of approved experimental protocols through the University of Pittsburgh IRB, and were compensated for their time.

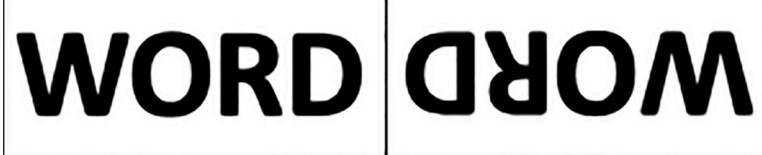

**Fig 1. Example of upright and inverted word presentations in lexical decision task.**

### Materials

**Lexical decision.** Sensitivity to atypical orientation was initially assessed using a lexical decision task in which the stimuli were presented in upright and inverted (rotated 180 degree, see Fig 1) orientations, the same atypical configuration used by Ben-Yehudah and colleagues [11]. Ben-Yehudah and colleagues used a naming paradigm, but here a lexical decision task was initially used to collect orientation sensitivity data without overt responses due to logistical constraints. Words were blocked such that upright stimuli were presented first, followed by inverted stimuli, with words and nonwords randomized within a block. Words were chosen to neither have extremely high nor low lexical frequency (min log HAL frequency = 6.27, max = 12.85, mean = 8.89). There were 20 words and 20 nonwords, with half of each presented in each orientation.

**Overt word naming.** Sensitivity to atypical orientation was confirmed using an overt naming task in which half of the stimuli were presented in typical orientation and half were presented in a reversed (FLIGHT → THGILF) orientation. Rather than use the same distortion manipulation as the lexical decision task, we visually distorted the words in a different way, and reason that similar group assignment provides a robust generalization of orientation sensitivity beyond inversion, and is consistent with diverse approaches used in the cross-linguistic literature. The stimuli were the 465 monosyllabic words used by Graves and colleagues [15] in a parametric neuroimaging study of word recognition. The items vary in length and along lexical (e.g., frequency and imageability), and sublexical (e.g., consistency, bigram frequency, biphone frequency) factors, and were selected to ensure that all factors are uncorrelated with each other within the stimulus list. The set of stimuli assigned to each orientation condition was matched along each of the dimensions sampled by Graves et al. [15].

**Table 1. Group statistics.**

| | | Mean | Std. Error | t | d | p |
|---|---|---|---|---|---|---|
| **Word Identification** | LS | 547 | 2.09 | 0.68 | 0.23 | 0.51 |
| | HS | 545 | 2.92 | | | |
| **Vocabulary** | LS | 14.43 | 0.47 | 0.78 | 0.31 | 0.44 |
| | HS | 13.82 | 0.66 | | | |
| **Spelling** | LS | 2.06 | 0.09 | -0.21 | 0.10 | 0.83 |
| | HS | 2.10 | 0.13 | | | |
| **Phonological Awareness** | LS | 102 | 3.34 | -0.25 | 0.09 | 0.81 |
| | HS | 103 | 2.46 | | | |
| **Phonemic Decoding** | LS | 521 | 1.88 | 0.52 | 0.29 | 0.61 |
| | HS | 519 | 2.05 | | | |
| **Comprehension** | LS | 539 | 1.80 | 1.79 | 0.69 | 0.09 |
| | HS | 534 | 2.32 | | | |

Independent samples t-tests were performed. Effect size is reported as Cohen's *d*.

**Reading skills.** The reading skills of participants were assessed using three subtests of the Woodcock Reading Mastery Tests [WRMT-Revised, Form H; 27]. The WRMT subtests included word identification (Word ID), in which subjects read aloud a list of words, phonemic decoding (Word Attack), in which subjects read aloud a list of nonwords, and Passage Comprehension, which requires subjects to supply missing words that best fit the context of short passages. Phonological awareness was assessed using the Comprehensive Test of Phonological Processing (CTOPP) subtests, Elision and Blending Words, that make up the Phonological Awareness Composite Score [28]. Vocabulary was assessed using the Wechsler Adult Intelligence Scale (WAIS-IV) Vocabulary Test [29]. Spelling was tested using the Lexical Knowledge Battery developed by Perfetti and Hart [21], which consisted of a list of 70 correctly and 70 incorrectly spelled real words.

## Procedure

Participants were tested individually in a quiet room during two sessions. The first session was a behavioral screening implemented in an online platform for a larger project. It included the Lexical Decision Task, which was used to identify potential participants for the current study, a Spelling Test, and Ravens Matrices [26]. In the Lexical Decision Task, participants made a two alternative forced decision about each stimulus (real word or nonword), presented in blocks (upright and inverted). The task was self-paced, with the next word appearing after each decision had been preceded by a fixation cross for 500ms. During the Spelling Test, participants saw an entire list of correctly and incorrectly spelled words, and were asked to select the ones that they were confident were spelled correctly. The remainder of this session was devoted to tasks for an ongoing database data collection, and it will not be discussed further. Participants were given a break after each task.

After participants were identified as eligible (see Group Definition Criteria above), additional testing took approximately two hours, including a mandatory 10-minute break halfway through the session. During the two-hour session, a sequence of tasks was administered in a predetermined constant order across all participants.

The WRMT subtests [27], WAIS Vocabulary Test, and CTOPP subtests were administered according to the published procedures. For each test, the experimenter gave detailed instructions and practice items, if applicable. All of the participants understood the instructions and completed the practice items successfully.

The overt naming task was administered using a Dell Dimension DIM4700 computer with a 17-inch screen. The stimuli were displayed using E-Prime software (Version 1.1, Psychology Software Tools, Inc., Pittsburgh, PA). The participants viewed the stimuli from a distance of approximately 50 cm. A voice key incorporated into a serial response box (Model 200A, Psychology Software Tools) recorded the time it took the participant to overtly pronounce each item from the moment it appeared on the screen (i.e., RT); accuracy was coded offline from a recording of the participant's overt responses. Participants read aloud words that were presented in a typical or atypical (reversed) orientation. The stimuli appeared at the center of the screen, in black lowercase letters against a white background.

The orientation (typical or reversed) of the displayed items was blocked; therefore, each item list was associated with either the typical or the reversed condition. Across participants, the order of the display condition was counterbalanced, such that half of the participants began with the typical orientation and the other half began with the reversed orientation. We chose to block the orientation of the stimuli to avoid task switching confounds [30].

Each block began with a cue indicating the orientation of the displayed items. Within each block, the trial began with a 500 ms black fixation-cross followed by the stimulus, which

appeared at the center of the screen and remained there until the participants responded. Following the overt response, the item was replaced by a fixation cross that cued participants to press a button when they were ready for the next trial. Within each block, items appeared in random order, without replacement. Participants were instructed to read the items presented aloud as quickly as they could without making any errors. Each participant completed practice trials before data collection commenced: 3 typical and 3 reversed items.

## Data analysis

**Reading skills.** For WRMT subtests, W-scores were used for statistical analyses. W-scores provide an equal-interval measure of test performance and they are the preferred measure for most statistical comparisons of group differences [27]. Raw scores for Elision and Blending Words were converted to standard scores, and then combined and converted to the Phonological Awareness Composite Score. Raw scores for WAIS Vocabulary were converted to scaled scores. Participant performance for the Spelling test was measured by d′ (i.e., sensitivity to misspelled words).

## Results

### Group statistics on reading skills

There were no significant differences in any component of reading skill between the two groups (Table 1). No measure even approached significance except comprehension.

### Factors affecting overt word naming

**Reaction time.** We used a linear mixed effects model to understand the impact of lexical and sublexical factors on overt word naming RTs. All incorrect trials were removed. The model included fixed effects of group (HS readers coded as 1, LS as 0), visual presentation orientation (typical = 0 or atypical = 1), lexical and sublexical factors (frequency, consistency, imageability, bigram frequency, biphone frequency, and length), and 2-way and 3-way interactions between group, visual presentation, and lexical/sublexical factors. All lexical and sublexical factors were mean-centered. The model also included random intercepts for individual words, participants, and the effect of lexical frequency across participants. The model was fit using the R software [31] and the lme4 package [32]. Of special interest were the 3-way interactions between group, visual word presentation, and each of the lexical and sublexical factors. A significant interaction would indicate that a particular lexical or sublexical factor has a larger influence on RT in one group over the other, depending on the orientation of the word. We hypothesized that the HS group should be more affected than the LS group by *lexical-level* factors when words are atypically oriented, whereas LS should be more affected than HS by *sublexical-level* factors when words are atypically oriented.

Results of all main effects, 2- and 3-way interactions are reported in Table 2. There were no significant interactions between Group and lexical/sublexical factors when looking at just typically presented words, but interactions with group emerged when looking at just atypically oriented words (see Fig 2). Most notably, there was a significant 3-way group x Presentation Orientation x Length interaction, such that the effect of length was relatively larger for HS readers when the words were presented in a reversed orientation. This suggests that HS are relatively slower for longer reversed words compared to LS readers. There were also two marginally significant 3-way interactions with Group (Group x Presentation Orientation x Imageability and Group x Presentation Orientation x Biphone Frequency). The 3-way interaction between Group, Presentation Orientation, and Imageability was such that HS were

**Table 2. Results of linear mixed model with reaction time as dependent variable.**

| | Estimate | Std. Error | t | P | |
|---|---|---|---|---|---|
| (Intercept) | 498 | 52.4 | 9.51 | < .001 | *** |
| Group | -47.0 | 78.7 | -0.60 | 0.56 | |
| Presentation Orientation | 210 | 6.95 | 30.2 | < .001 | *** |
| Frequency | -35.6 | 9.44 | -3.77 | 0.00 | *** |
| Imageability | -13.2 | 5.38 | -2.45 | 0.01 | * |
| Length | 20.3 | 8.86 | 2.29 | 0.02 | * |
| Consistency | -3.18 | 0.78 | -4.10 | < .001 | *** |
| Bigram | -1.66 | 12.8 | -0.13 | 0.90 | |
| Biphone | -1.77 | 1.18 | -1.51 | 0.13 | |
| Group x Presentation Orientation | 268 | 10.5 | 25.4 | < .001 | *** |
| Group x Frequency | -10.2 | 11.8 | -0.87 | 0.39 | |
| Group x Imageability | 0.21 | 6.08 | 0.03 | 0.97 | |
| Group x Length | -9.49 | 10.0 | -0.95 | 0.34 | |
| Group x Consistency | -0.28 | 0.87 | -0.33 | 0.74 | |
| Group x Bigram | 14.7 | 14.4 | 1.02 | 0.31 | |
| Group x Biphone | -0.33 | 1.33 | -0.25 | 0.80 | |
| Presentation Orientation x Frequency | -17.8 | 8.48 | -2.10 | 0.04 | * |
| Presentation Orientation x Imageability | -8.82 | 5.78 | -1.53 | 0.13 | |
| Presentation Orientation x Length | 74.6 | 9.45 | 7.90 | < .001 | *** |
| Presentation Orientation x Consistency | -0.28 | 0.83 | -0.34 | 0.74 | |
| Presentation Orientation x Bigram | -5.54 | 13.7 | -0.41 | 0.69 | |
| Presentation Orientation x Biphone | -1.28 | 1.25 | -1.02 | 0.31 | |
| Group x Presentation Orientation x Frequency | 4.05 | 12.9 | 0.32 | 0.75 | |
| Group x Presentation Orientation x Imageability | -16.9 | 8.75 | -1.93 | 0.054 | . |
| Group x Presentation Orientation x Length | 55.7 | 14.3 | 3.89 | < .001 | *** |
| Group x Presentation Orientation x Consistency | 0.89 | 1.25 | 0.71 | 0.48 | |
| Group x Presentation Orientation x Bigram | -24.5 | 20.7 | -1.18 | 0.24 | |
| Group x Presentation Orientation x Biphone | 3.63 | 1.90 | 1.91 | 0.056 | . |

*** = $p < .001$
* = $p < .05$, . = $p < .10$
Effects involving lexical factors are highlighted in gray, and effects involving sublexical factors have a white background.

relatively faster to name words that were imageable when they were presented in a reversed orientation. The opposite pattern is seen with the interaction between Group, Presentation Orientation, and Biphone Frequency, such that it was the LS readers who were faster to name words that had higher biphone frequency, when words were presented in a reversed orientation. A power analysis [33] suited to linear mixed models [34] was run, which estimates the power for specific effects in our model using Monte Carlo estimation. This was conducted for each of the three 3-way interactions that showed effects, set at 200 simulations. The 3-way interaction between group x orientation x length had a very large effect size, and the observed power was 96.50% (95% confidence interval: 92.92, 98.58). The two marginal effects both had 51% power (43.85, 58.12).

An additional model was run to examine a potential concern that group selection should not be based on the data run in this analysis. The new model additionally included the previously removed participants and coded "group" as a continuous variable based solely on the lexical decision task (LDT) and not the overt naming data. The 3-way interaction between

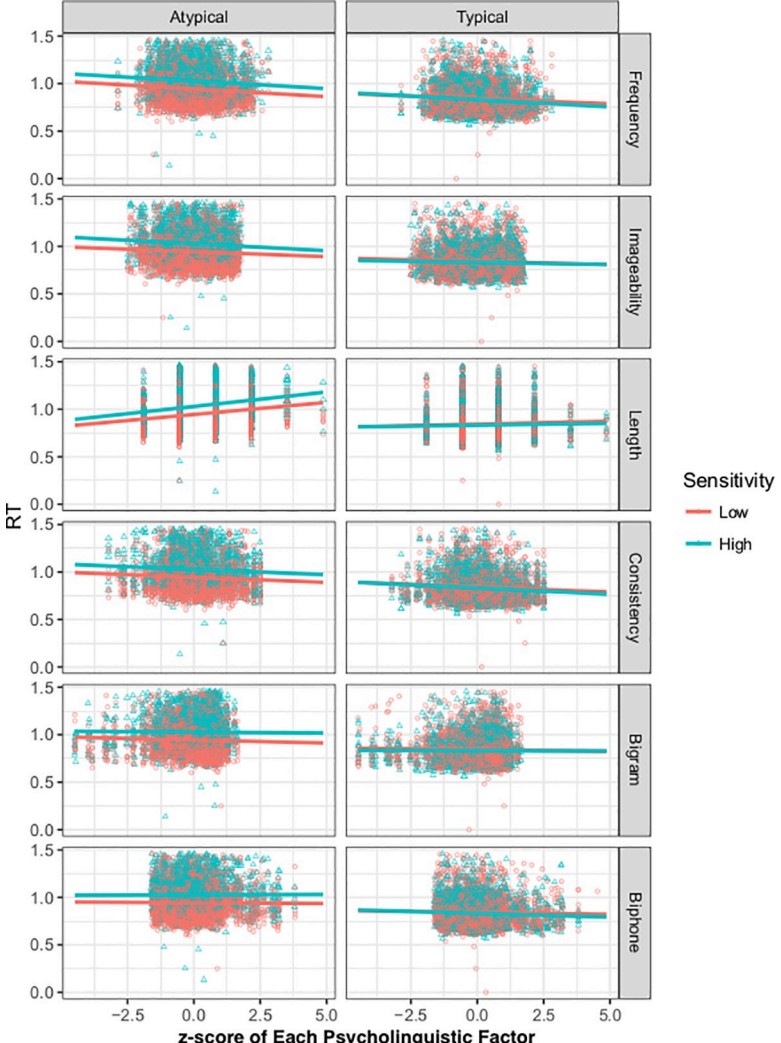

**Fig 2. Reaction Time for Group X Presentation Orientation X Psycholinguistic Factor 3-way interactions.** Data for typical and atypical word presentations are graphed separately. Data points are reaction times from individual trials pooled across participants with outliers (± 2 standard deviation) removed.

"group" (i.e., strength of inversion sensitivity in LDT task) x presentation orientation x length remained highly significant and even stronger (t = 6.12, p < .001). The two marginal 3-way interactions (between group x orientation x imageability and biphone frequency) were no longer significant in the new model, but another marginal 3-way interaction emerged in the predicted direction (t = 1.79, p = .07), such that bigram frequency was a better predictor of RT for atypically presented words in those whose LDT were relatively small (i.e., those with low sensitivity).

**Accuracy.** Naming accuracy was close to ceiling (LS: M = .96, SD = .20; HS: M = .93, SD = .25), which posed problems for convergence of a general linear mixed effects model (glmer). Therefore, a secondary analysis was conducted using a weighted empirical logit model, which is designed for cases of near floor or ceiling performances (i.e., the accuracy probability across subjects is near zero or one) [see 35]. An 'empirical log odds' of accuracy (log (correct trials + .5/incorrect trials + .5) was computed for each word in one of four bins, crossing group x

presentation orientation (i.e., HS typical words, HS atypical words, LS typical words, LS atypical words). Each value was also weighted to account for a different number of total trials for a given word in a given bin, due to unequal sample size in each group, etc. The model was then essentially the same as the reaction time model, with the exception that there was no random effect of subject.

Results of all main effects, 2- and 3-way interactions are reported in Table 3. Of note were a significant 2-way interestion between Group X Biphone Frequency and marginally significant 3-way interaction between Group x Presentation Orientation x Biphone Frequency. In both, LS has a larger positive relationship between accuracy and biphone frequency. The marginally significant 3-way interaction suggests that this effect is slightly amplified when words were typically presented words.

### Principal component analyses

**Lexical representational structure.**   Recognizing that group numbers are small, exploratory factor analyses were computed in each group separately [36]. Lexical representational structure was assessed using a principal component analysis (PCA) in each group separately to characterize the structure of the relationship between Word ID and three tasks that emphasize knowledge of lexical constituents (i.e., spelling, which assesses orthography; phonological awareness, which assesses phonology; vocabulary, which assesses semantics), and phonemic decoding (which assesses knowledge of the correspondence between orthography and phonology). PCA using a Varimax rotation with a Kaiser normalization was employed. Only eigenvalues greater than one were considered for component identification.

In LS, the PCA identified a single component, accounting for 55.06% of the variance. In contrast, the PCA in HS identified two components that accounted for 78.43% of the variance (49.77% by the first, and 33.92% by the second) (Table 4). The first component had high factor loadings for Word ID, spelling, vocabulary (.898, .814, and .886 respectively), medium factor loading for phonological awareness (.452), and low for phonemic decoding (-.175). The second component had high factor loadings for phonological awareness and phonemic decoding (.858, .897).

Due to the differences in variance explained in the two PCAs, we then ran an additional analysis with LS that forced a two factor solution. The two components together accounted for 73.08% of the variance. Interestingly, Word ID (.935), phonological awareness (.750) and phonemic decoding (.728) loaded more heavily and together on the first component (Table 5). This is in contrast to the HS, where Word ID and the two phonological measures strongly loaded on separate components, with phonemic decoding specifically only loading on the second component, suggesting that it is less correlated with Word ID in HS (Fig 3). A follow-up analysis revealed that phonemic decoding was more correlated with Word ID in LS ($r$ = .65, $p$ = .013) than HS ($r$ = .03, $p$ = .84) in separate regressions for each group, although a group x phonemic decoding interaction in a combined model did not explain a significant amount of additional variance ($p$ = .19).

### Discussion

The present study examined whether differences in holistic orthographic coding, measured by sensitivity to orientation, predict differences in the reading procedures of native English readers. More specifically, orientation sensitivity was hypothesized to occur with a bias towards lexical reading procedures, which should be an indicator of a reading profile that has less reliance on sublexical processing and phonological decoding. While only one effect was highly significant, all effects including marginal effects demonstrated this overall pattern when looking at

**Table 3. Results of weighted empirical logit model with accuracy log odds as dependent variable.**

| | Estimate | Std. Error | t | p | |
|---|---|---|---|---|---|
| (Intercept) | 2.41 | 0.04 | 59.51 | < .001 | *** |
| Group | -0.28 | 0.04 | -6.30 | < .001 | *** |
| Presentation Orientation | -0.19 | 0.04 | -4.51 | < .001 | *** |
| Frequency | 0.21 | 0.05 | 4.35 | 0.00 | *** |
| Imageability | 0.09 | 0.03 | 2.72 | 0.01 | ** |
| Length | -0.02 | 0.06 | -0.36 | 0.72 | |
| Consistency | 0.01 | 0.00 | 1.58 | 0.11 | |
| Bigram | -0.10 | 0.08 | -1.30 | 0.19 | |
| Biphone | 0.02 | 0.01 | 2.79 | 0.01 | ** |
| Group x Presentation Orientation | -0.23 | 0.06 | -3.77 | < .001 | *** |
| Group x Frequency | -0.01 | 0.05 | -0.19 | 0.85 | |
| Group x Imageability | 0.00 | 0.04 | -0.04 | 0.96 | |
| Group x Length | 0.02 | 0.06 | 0.31 | 0.75 | |
| Group x Consistency | 0.00 | 0.01 | 0.18 | 0.85 | |
| Group x Bigram | -0.07 | 0.09 | -0.79 | 0.43 | |
| Group x Biphone | -0.02 | 0.01 | -2.06 | 0.04 | * |
| Presentation Orientation x Frequency | -0.04 | 0.05 | -0.74 | 0.46 | |
| Presentation Orientation x Imageability | -0.02 | 0.03 | -0.67 | 0.51 | |
| Presentation Orientation x Length | -0.07 | 0.06 | -1.22 | 0.22 | |
| Presentation Orientation x Consistency | 0.01 | 0.00 | 1.52 | 0.13 | |
| Presentation Orientation x Bigram | 0.03 | 0.09 | 0.31 | 0.76 | |
| Presentation Orientation x Biphone | -0.01 | 0.01 | -1.59 | 0.11 | |
| Group x Presentation Orientation x Frequency | 0.11 | 0.07 | 1.58 | 0.11 | |
| Group x Presentation Orientation x Imageability | 0.02 | 0.05 | 0.42 | 0.67 | |
| Group x Presentation Orientation x Length | -0.11 | 0.08 | -1.31 | 0.19 | |
| Group x Presentation Orientation x Consistency | -0.01 | 0.01 | -1.27 | 0.20 | |
| Group x Presentation Orientation x Bigram | 0.03 | 0.13 | 0.25 | 0.80 | |
| Group x Presentation Orientation x Biphone | 0.02 | 0.01 | 1.85 | 0.06 | . |

*** = p < .001
* = p < .05,. = p < .10
Effects involving lexical factors are highlighted in gray, and effects involving sublexical factors have a white background.

**Table 4. Factor loadings for lexical representational structure.**

| | Component | | Component | |
|---|---|---|---|---|
| **Lower Sensitivity Readers** | **1** | **Higher Sensitivity Readers** | **1** | **2** |
| Word Identification | 0.894 | Word Identification | 0.898 | 0.25 |
| Vocabulary | 0.823 | Vocabulary | 0.886 | 0.149 |
| Spelling | 0.487 | Spelling | 0.814 | -0.266 |
| Phonological Awareness | 0.759 | Phonological Awareness | 0.452 | 0.858 |
| Phonemic Decoding | 0.681 | Phonemic Decoding | -0.175 | 0.897 |

Variables highlighted in gray denote lexical factors. Variables highlighted in white denote sublexical factors.

**Table 5. Factor Loadings for a forced 2-component model of lexical representational structure for lower sensitivity readers.**

| | Component | |
|---|---|---|
| **Lower Sensitivity Readers** | **1** | **2** |
| Word Identification | 0.411 | 0.816 |
| Vocabulary | 0.827 | 0.225 |
| Spelling | 0.689 | 0.083 |
| Phonological Awareness | 0.714 | 0.204 |
| Phonemic Decoding | 0.071 | 0.935 |

Variables highlighted in gray denote lexical factors. Variables highlighted in white denote sublexical factors.

predictors of overt word naming reaction time and accuracy, as well as exploratory factor analyses examining lexical structural representation. Thus, while individual results must be regarded with caution due to the small sample size and marginal significance, they cohere together as predicted, and suggest that orientation sensitivity can be used as a marker of reading procedures and to unmask reading procedure differences in highly skilled readers.

The results from the overt naming task are consistent with our hypotheses and past studies looking at Chinese-English and Koren-English bilinguals, which found that greater orientation sensitivity covaries with a bias towards lexical reading procedures [11, 12]. Conversely, less orientation sensitivity covaries with a bias towards sublexical reading procedures. The current study expanded the scope of previous studies with a more thorough contrast of lexical vs. sublexical factors using a word list controlled for additional psycholinguistic factors [e.g., imageability, biphone frequency, etc.; 15]. The 3-way interactions (marginally significant) between Group x Presentation Orientation x Biphone Frequency in both reaction time and accuracy supported this hypothesis. LS individuals were more affected by biphone frequency than HS individuals in the reversed orientation condition (i.e., relatively slower RTs and lower accuracy for atypically oriented words with lower biphone frequency). While the marginally significant 3-way interaction with biphone frequency in the reaction time data did not hold in a follow-up analysis, a 3-way interaction with bigram frequency emerged in the same and predicted direction, supporting the same, albeit weak, overall pattern. Overall, this pattern is more similar to Korean-English bilingual performance in past studies and is consistent with a more analytical/sublexical reading procedure.

In addition to HS relying less on a sublexical factor compared to LS, there was some evidence that they also relied more on lexical-level factors. First, there was a marginally significant 3-way interaction between Group x Presentation Orientation x Imageability in reaction time data. Heightened sensitivity to imageability was a predicted outcome for individuals with a bias towards using lexical reading procedures since imageability can only be assessed at the whole-word level. Second, there was a highly significant 3-way interaction between Group x Presentation Orientation x Length in reaction time data, such that HS had relatively longer reaction times for longer words when they were atypically oriented. This pattern was robust to an alternative reanalysis. Further, a power analysis and recent simulations exploring issues of replicability suggest this is unlikely to be a spurious result, even when the small sample size is taken into account [37]. The significant 2-way interaction between presentation orientation and length provides support for the idea that atypical orientation leads to the requirement for a more effortful sublexical approach in word identification, thus leading to longer reading times for longer words [6, 7]. The fact that the length effect for atypically oriented words was larger for HS suggests that they were less efficient at utilizing the sublexical/analytical

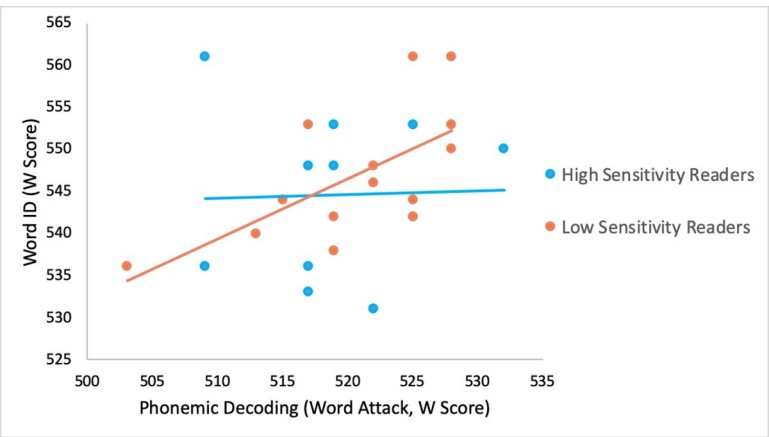

**Fig 3. Correlation between word ID and phonemic decoding in low and high sensitivity readers.**

approach, and thus had relatively longer reaction times than LS for longer words when they were atypically oriented. This pattern is more similar to Chinese-English bilingual performance in past studies and is consistent with a more holistic/lexical reading procedure.

These behavioral patterns observed in HS readers are also consistent with recent fMRI results, examining a subset of the same participants in the current study [25]. Greater orientation sensitivity was associated with bilateral visual word form area (VWFA) engagement in the mid-fusiform gyrus (mFG). This pattern is also consistent with artificial orthography studies that suggest attention to sublexical decoding is linked with left hemisphere dominant mFG activity [38], whereas decoding using larger grain sizes is associated with relatively more bilateral mFG activity [39–41].

The exploratory factor analysis of lexical structural representation, including tasks that emphasize lexical constituent knowledge (i.e., spelling, phonological awareness and vocabulary) and phonemic decoding, revealed a single component for LS readers. In contrast, for HS readers, the measures loaded on two components, with spelling and vocabulary loading more on a first component with word identification, and phonemic decoding loading more on a second component. Phonological awareness was more balanced than the other factors, but had a stronger loading on the second component, with phonemic decoding. These results suggest that the two groups' lexical representations may be structured differently (see Tables 4 & 5). When the models were run without restrictions, LS readers have a more cohesive structure of all factors, with all measures weighing on a single component. In contrast, for HS readers, phonological measures, especially phonemic decoding, weigh more heavily on a second component, separate from a measure of word identification, spelling, and vocabulary. In contrast, when the LS model was forced to have two components, phonological measures and word identification loaded heavily on the same component.

These exploratory factor analysis results are intriguing because the structure of lexical representations has been linked with overall reading skill. The Lexical Quality Hypothesis (Perfetti & Hart, 2001; 2002) proposes that the quality of the lexical representation can be measured by the degree to which all factors are highly redundant or correlated, leading to specific, coherent, and reliable word identification. The constituents of skilled readers tend to load on one or two components in a factor analysis, while less skilled readers' constituents load on more components, indicating a less cohesive representation. What is interesting about the current exploratory results is that the coherence of the two groups' lexical structural representations differed even though the groups were matched along measures of reading skill. Thus, the results suggest

that less coherent structure in one's lexical representation does not necessarily result in less skilled reading, in turn raising questions about whether more than one profile of lexical integration can support skilled reading.

Taken together, the HS profile is similar to previously observed differences amongst small subpopulations of English readers. There are documented subgroups of readers that similarly have shown high levels of comprehension with a weaker link between comprehension and phonological decoding: resilient readers [42, 43] and deaf native signers [44–47]. However, these subgroups differ from our HS group in that they have lower levels of phonological decoding, leading to the inference that these groups use lexical procedures to 'compensate for' their poor phonology skill and thereby achieve high levels of literacy. However, one could argue that our HS readers (and previously reported Chinese-English readers) did not need to compensate for poor phonological processing, since they have normal ranges of phonological skills. Future research is required to assess whether orientation sensitivity would distinguish skilled resilient readers and deaf native signers from appropriate control groups matched for component skills (e.g., phonological decoding, etc.), but with poor comprehension.

In future work, it will be important to determine whether the results observed in this study generalize to other samples and other orientation manipulations. There is ongoing debate, especially in the face processing literature [48], about which tasks encompass the construct of 'holistic' processing. A general consensus is that various tasks (e.g., inversion, composite task [10, 49]) have some unique variance, highlighting the importance of using a combination of measures, including a composite task. An open question also remains as to what leads to variation in orientation sensitivity in skilled English readers. Based on our initial sample, higher sensitivity to atypical orientation (i.e., a ratio of above 1.5 on the lexical decision task) is fairly common, although more research is needed to assess how prevalent it is in a wider population and how that relates to reading procedures. It is possible that we are simply observing a natural variation in cognitive biases present in the population. Alternatively, some variation could result from differences in foundational instructional methods. There is evidence for differences in some reading procedures (e.g., nonword decoding) in children who were taught with a phonics approach vs. a story/text-centered approach that focuses on context cues and analogies rather than sounding words out [8, 50]. Future research is needed to determine whether the ability to manipulate reading biases through instruction could benefit individuals with poor phonological decoding who did not naturally rely on a reading procedure that best fits their cognitive abilities.

## Supporting information

**S1 Data.**
(CSV)

**S2 Data.**
(CSV)

## Acknowledgments

The authors would like to thank Scott Fraundorf and Ting Qian for their assistance with the linear mixed modeling analyses.

## Author Contributions

**Conceptualization:** Elizabeth A. Hirshorn, Travis Simcox, Charles A. Perfetti, Julie A. Fiez.

**Data curation:** Elizabeth A. Hirshorn, Corrine Durisko.

**Formal analysis:** Elizabeth A. Hirshorn.

**Funding acquisition:** Julie A. Fiez.

**Investigation:** Elizabeth A. Hirshorn, Travis Simcox, Corrine Durisko, Julie A. Fiez.

**Methodology:** Travis Simcox, Corrine Durisko, Julie A. Fiez.

**Project administration:** Travis Simcox, Corrine Durisko, Julie A. Fiez.

**Supervision:** Julie A. Fiez.

**Writing – original draft:** Elizabeth A. Hirshorn.

**Writing – review & editing:** Elizabeth A. Hirshorn, Charles A. Perfetti, Julie A. Fiez.

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
