## [Decision Letter · Decision Letter 0]

25 Sep 2019

PONE-D-19-17391

Unmasking individual differences in adult reading procedures by disrupting holistic orthographic perception

PLOS ONE

Dear Dr. Hirshorn,

Thank you for submitting your manuscript to PLOS ONE. After careful consideration, we feel that it has merit but does not fully meet PLOS ONE’s publication criteria as it currently stands. Therefore, we invite you to submit a revised version of the manuscript that addresses the points raised during the review process.

I have sent your manuscript to three external reviewers. You will see below that there were several concerns raised in the course of the review that led to the decision that the research article is not acceptable in its current form. Reviewers raised some concerns about the small sample size and about the experimental design and procedure. I therefore invite you to revise and resubmit your manuscript, taking into account the points raised by the reviewers. Please also consider the other comments of the reviewers as you prepare your revision. 

We would appreciate receiving your revised manuscript by Nov 09 2019 11:59PM. To enhance the reproducibility of your results, we recommend that if applicable you deposit your laboratory protocols in protocols.io, where a protocol can be assigned its own identifier (DOI) such that it can be cited independently in the future. For instructions see: http://journals.plos.org/plosone/s/submission-guidelines#loc-laboratory-protocols

We look forward to receiving your revised manuscript.

Kind regards,

Yafit Gabay

Academic Editor

PLOS ONE

Journal Requirements:

Reviewers' comments:

Reviewer's Responses to Questions

**Comments to the Author**

1. Is the manuscript technically sound, and do the data support the conclusions?

Reviewer #1: Partly

Reviewer #2: Partly

Reviewer #3: Partly

2. Has the statistical analysis been performed appropriately and rigorously? 

Reviewer #1: I Don't Know

Reviewer #2: Yes

Reviewer #3: No

3. Have the authors made all data underlying the findings in their manuscript fully available?

Reviewer #1: Yes

Reviewer #2: Yes

Reviewer #3: Yes

4. Is the manuscript presented in an intelligible fashion and written in standard English?

Reviewer #1: Yes

Reviewer #2: Yes

Reviewer #3: Yes

5. Review Comments to the Author

Reviewer #1: OVERALL IMPRESSION:

My overall impression is somewhat positive re. the study overall but I am worried about the possibility of low power. I think that the main idea is interesting, and it seems that the study was well executed. Sample size is small (N = 14 and N = 11) which made me question the replicability of certain outcomes, both positive and negative. In any case, a small sample makes it particularly important IMO to show the underlying data points explicitly and graphically in the manuscript.

COMMENTS AND SUGGESTIONS:

-- Introduction --

“An atypical orientation is thought to disproportionately affect holistically processed objects because the holistic process cannot be applied when the object is presented unconventionally.” While the former part is widely accepted, I wanted to point the authors to this counterargument to the latter part:

Richler, J. J., Mack, M. L., Palmeri, T. J., & Gauthier, I. (2011). Inverted faces are (eventually) processed holistically. Vision Research, 51(3), 333-342.

“Chinese is a morpho-syllabic writing system, and so sublexical orthographic coding and mapping to phonology is less useful for word identification, as compared to the holistic coding and mapping of characters to their morphemic forms.” Whether or not expert Chinese readers rely on holistic processing of Chinese characters might depend on their writing experience: “Compared with Chinese nonreaders, Chinese readers who had limited writing experience showed increased holistic processing, whereas Chinese readers who could write characters fluently showed reduced holistic processing.”

Tso, R. V. Y., Au, T. K. F., & Hsiao, J. H. W. (2014). Perceptual expertise: can sensorimotor experience change holistic processing and left-side bias? Psychological Science, 25(9), 1757-1767.

-- Materials and Methods –

---- Lexical decision and Overt word naming----

As already mentioned, the sample size was quite small. The initial 2AFC (I am assuming, not much info is given – I suggest giving more detail about tasks and stimuli, randomization/blocking etc.) lexical decision screening task also only included 40 trials, which might lead to unreliable RT estimates and therefore unreliable group membership assignment.

I was therefore happy to see that the author reassessed orientation sensitivity. This manipulation is however different from the one used in lexical decision. In lexical decision, parts and wholes were both inverted (akin to the manipulation for the face inversion effect), while in overt word naming, the whole was disturbed while parts (i.e. letters) were kept in their original orientation. The authors however provide no explanation of why they used this other way of experimentally defining orientation sensitivity.

-- Results –

---- Table 1 ----

I suggest including Cohen’s d effect size estimates. Provide units (e.g. ms). I think that at least some of the stats might be wrong. E.g. I calculated Comprehension using https://www.graphpad.com/quickcalcs/ttest2/ I came up with a similar p-value (0.10 instead of 0.09) but a completely different t-value (1.73 compared to 0.51). More generally, please say explicitly in the table heading whether you are presenting regular t-tests or something else in that table.

---- Table 2, table 3, and figure 2 ----

I suggest clearly marking factors as lexical and sublexical as this was a major part of your hypothesis.

In order to interpret certain factors in the models, the reader needs to know how they were coded. Was e.g. LS 0 and HS 1? Same with others, e.g. presentation orientation.

What would help even more with interpretation of these, frankly, quite complicated models with several factors and multiple interactions, is if you would actually present the underlying data graphically. This is done in figure 2 for a few of the variables, but I strongly suggest that you show correlations in a different way that highlights the underlying data points more explicitly (e.g. scatterplots, correlograms: https://www.r-graph-gallery.com/correlogram/). This makes it much easier to compare the groups, and for scatterplots can simultaneously show overall group differences (group main effects) as well as interactions (e.g. to show that the association between orientation and RT is different for the groups). I also suggest that this should not only be shown for the significant interactions, as you expected other differences as well that did not come out as significantly different between the two groups.

---- Principal components analysis ----

The authors rightfully point out that the groups are small for an exploratory factor analysis. I have seen estimates of at least 50 participants, or of at least 10 times the number of participants as there are variables. I found one paper that claims that exploratory factor analysis can be done for very small sample sizes under certain circumstances:

de Winter*, J. D., Dodou*, D. I. M. I. T. R. A., & Wieringa, P. A. (2009). Exploratory factor analysis with small sample sizes. Multivariate behavioral research, 44(2), 147-181.

However, I think that given the small sample sizes, any differences in factor structure might be uninterpretable. If the authors want to show possible group differences as an exploratory or descriptive analysis, I again suggest that showing two correlograms, one per group, would be more likely to give the reader a gist of what might be going on.

Reviewer #2: This paper presents the results of a study that examined the existence of different reading styles in English. The majority of studies have been done on reading in English, and the consensual model, the dual route model, holds that written words can be identified via the lexical, or holistic/orthographic route, or a sub-lexical, phonological decoding route. A number of cross language/writing system studies have shown that the orthographic transparency and morphological structure of specific writing systems result in more efficient processing weighting the lexical or the sub-lexical route. In English, it has been shown that both characteristics of the words and of the reader (e.g., skill level) affect the relative use of the lexical or sublexical routes. I have several major concerns:

1. I am not exactly sure of the goal of this paper: participants were divided into a ‘holistic’ or ‘sub-lexical’ category by the degree to which reading words upside-down differed from reading words in the canonical orientation. Then, after a very extensive sifting procedure, these groups were compared on reading words forwards and backwards. I understand that the point was to see if performance on this task, using words for which the holistic and sub-lexical aspects are documented, would go in the same direction. Here lies the problem – the fact that the groups differed on the backwards reading task is trivial, as they were chosen to differ on an upside-down reading task. The fact that they only differ on performance on the upside-down words is problematic, as it suggests that in normal word identification tasks, they do not differ. Thus, when the task is made more difficult in a specific manner, then the groups differ in how well they can compensate for the spatial distortion. But that is how the groups were created to begin with – there is no theoretical reason to think that the two spatial manipulations (upside down and backwards) are inherently different.

2. The finding that the factor structure of the RTs is different in a predictable manner could be even more supporting evidence that this division is reasonable, but the very small number of participants in the groups (11 and 14) really makes it hard to believe it…-- and, it is also true by the way that the groups were defined.

3. There seem to be some missed opportunities here. Although it is not mentioned, the division of word characteristics into lexical and sub-lexical categories follows results shown by many studies of reading by split-brain patients, reading by unilateral brain damaged patients, and healthy participants identifying words in divided visual field paradigms. This has been done across languages (e.g., Rao & Vaid, 2017; Zhou et al, 2019; Ibrahim & Eviatar, 2012; and many more). Given the very large initial sample, it would have been interesting to see if left handed participants tended to fall into one or the other group. That would have been a novel result.

Summary: Other than the very small sample size, the study seems well done. I am just not sure what it adds to the literature.

Reviewer #3: The investigators have concluded that their results revealed that greater orientation sensitivity was associated with a reading profile that relies less on sublexical phonological measures and more on lexical-level characteristics within the skilled English readers. This is based on a statistical approach involving typical procedures such as the t-test, mixed models, the empirical logit model and principal component analysis. There are some concerns. Specifically:

1. The authors note that participants were recruited from a database of 411 individuals interested in study opportunities. It is not clear what ‘study opportunities’ means here or if it is even relevant. They explain that a number of subjects were removed leaving 203 from an initial 411 potential participants. This appears to be a convenience sample. A hypothesis is stated on page 7 in the form that native English readers who show more sensitivity to atypical orientation do so because they make greater use of holistic orthographic coding, which should be reflected in a distinctive reading profile. This appears to be a comparison of the low to high sensitivity group which they define. The problem is that there is no statistical design motivation for the sample size. Is 203 individuals statistically sufficient to test their hypothesis with reasonable statistical power? Also on the bottom of Page 8 the investigators note that participant attrition, due to graduation since participation in initial screening and eligibility requirements for a parallel imaging study (beyond the scope of the current study), also reduced the potential participant pool. What exactly does this mean and what was the final number of participants for each of the Tables 1 to 5?

2. As minor points please define RT on page 8. It appears to be 'reaction time'. On the top line of page 20 the term ‘ phonological decoding (.728)’ should be ‘Phonemic Decoding (0.728)’ as per Table 5.

6. PLOS authors have the option to publish the peer review history of their article (what does this mean?). If published, this will include your full peer review and any attached files.

Reviewer #1: No

Reviewer #2: No

Reviewer #3: No

---

## [Author Response · Author response to Decision Letter 0]

21 Nov 2019

Dear Dr. Yafit Gabay and Reviewers,

We very much appreciate the helpful comments and believe they have greatly improved the paper. Responses to individual comments are below and also uploaded as "ResponsetoReviewers_Final.docx".

Sincerely, 

Elizabeth Hirshorn, PhD

Reviewer #1: OVERALL IMPRESSION:

My overall impression is somewhat positive re. the study overall but I am worried about the possibility of low power. I think that the main idea is interesting, and it seems that the study was well executed. Sample size is small (N = 14 and N = 11) which made me question the replicability of certain outcomes, both positive and negative. In any case, a small sample makes it particularly important IMO to show the underlying data points explicitly and graphically in the manuscript.

We appreciate this point and have included figures for the most critical results of the paper (see new Fig 2 and 3). We understand the concern of small sample size and have made a point to be transparent in describing strong results that are statistically unlikely due to chance, and versus those that are more tentative. We have also added text to our discussion section (pgs. 23 and 27) that explicitly notes our relatively low sample size and that recommends future work to test whether the results generalize to other groups.

COMMENTS AND SUGGESTIONS:

-- Introduction --

“An atypical orientation is thought to disproportionately affect holistically processed objects because the holistic process cannot be applied when the object is presented unconventionally.” While the former part is widely accepted, I wanted to point the authors to this counterargument to the latter part:

Richler, J. J., Mack, M. L., Palmeri, T. J., & Gauthier, I. (2011). Inverted faces are (eventually) processed holistically. Vision Research, 51(3), 333-342.

Thank you for this point. We have added this reference to the introduction at the bottom of page 3 of the manuscript. We would argue that even if perception is ultimately holistic, inversion sensitivity could still be used as a marker of reading procedures.

“Chinese is a morpho-syllabic writing system, and so sublexical orthographic coding and mapping to phonology is less useful for word identification, as compared to the holistic coding and mapping of characters to their morphemic forms.” Whether or not expert Chinese readers rely on holistic processing of Chinese characters might depend on their writing experience: “Compared with Chinese nonreaders, Chinese readers who had limited writing experience showed increased holistic processing, whereas Chinese readers who could write characters fluently showed reduced holistic processing.”

Tso, R. V. Y., Au, T. K. F., & Hsiao, J. H. W. (2014). Perceptual expertise: can sensorimotor experience change holistic processing and left-side bias? Psychological Science, 25(9), 1757-1767.

This point is well taken. We have added this reference and made the point that the degree of holistic processing varies in Chinese readers, although Chinese readers may still on average rely more on holistic processing than alphabetic readers (see pg. 4-5 of manuscript).

-- Materials and Methods –

---- Lexical decision and Overt word naming----

As already mentioned, the sample size was quite small. The initial 2AFC (I am assuming, not much info is given – I suggest giving more detail about tasks and stimuli, randomization/blocking etc.) lexical decision screening task also only included 40 trials, which might lead to unreliable RT estimates and therefore unreliable group membership assignment.

This is correct. The initial lexical decision task was part of a larger screening battery, and therefore was meant to be very short. (In subsequent research, we have increased the number of trials.) More detail has been added to the methods on page 10 of the manuscript.

I was therefore happy to see that the author reassessed orientation sensitivity. This manipulation is however different from the one used in lexical decision. In lexical decision, parts and wholes were both inverted (akin to the manipulation for the face inversion effect), while in overt word naming, the whole was disturbed while parts (i.e. letters) were kept in their original orientation. The authors however provide no explanation of why they used this other way of experimentally defining orientation sensitivity.

Thank you for this comment, as we believe this information is important to convey clearly. By not using the same manipulation, we were able to generalize the orientation sensitivity of our groups. Other studies have used alternating font, etc., but we chose the reversed presentation because it was a larger distortion of the whole, as you point out, and would be pushing our hypotheses further. This is now communicated in the text on pg. 10.

-- Results –

---- Table 1 ----

I suggest including Cohen’s d effect size estimates. Provide units (e.g. ms). I think that at least some of the stats might be wrong. E.g. I calculated Comprehension using https://www.graphpad.com/quickcalcs/ttest2/ I came up with a similar p-value (0.10 instead of 0.09) but a completely different t-value (1.73 compared to 0.51). More generally, please say explicitly in the table heading whether you are presenting regular t-tests or something else in that table.

Thank you for this comment. Cohen’s d values have been added to Table 1 on pg. 14. Thank you for catching that typo for the t-value for the comprehension measure. We see now that it didn’t make much sense. Yes, I believe the p-value was just a rounding error, as it was .096. It is now reported as .10.

---- Table 2, table 3, and figure 2 ----

I suggest clearly marking factors as lexical and sublexical as this was a major part of your hypothesis.

Thank you for this helpful suggestion. It makes a lot of sense for clarity. In the tables (2 & 3), the effects involving lexical factors are now highlighted in light gray, and the sublexical factors have a white background. This is explained in the table footnotes.

In order to interpret certain factors in the models, the reader needs to know how they were coded. Was e.g. LS 0 and HS 1? Same with others, e.g. presentation orientation.

Point taken. Yes, LS was coded as 0, and upright/typically presented words were coded as 0. This is now explicitly noted in the text on pg. 14. 

What would help even more with interpretation of these, frankly, quite complicated models with several factors and multiple interactions, is if you would actually present the underlying data graphically. This is done in figure 2 for a few of the variables, but I strongly suggest that you show correlations in a different way that highlights the underlying data points more explicitly (e.g. scatterplots, correlograms: https://www.r-graph-gallery.com/correlogram/). This makes it much easier to compare the groups, and for scatterplots can simultaneously show overall group differences (group main effects) as well as interactions (e.g. to show that the association between orientation and RT is different for the groups). I also suggest that this should not only be shown for the significant interactions, as you expected other differences as well that did not come out as significantly different between the two groups.

Thank you for this suggestion. We believe it will aid in the transparency in understanding our results. We have included scatterplot figures for the 3-way interactions of the RT data, since that is where our hypotheses were more relevant. We presented the 3-way interactions as two scatterplots, one for typical and one for atypical (reversed) stimuli, with the x-axis representing the psycholinguistic factor (e.g. frequency) and the different colored dots/lines representing the groups. We included these for all psycholinguistic factors, even non-significant ones (see pg. 18).

---- Principal components analysis ----

The authors rightfully point out that the groups are small for an exploratory factor analysis. I have seen estimates of at least 50 participants, or of at least 10 times the number of participants as there are variables. I found one paper that claims that exploratory factor analysis can be done for very small sample sizes under certain circumstances:

de Winter*, J. D., Dodou*, D. I. M. I. T. R. A., & Wieringa, P. A. (2009). Exploratory factor analysis with small sample sizes. Multivariate behavioral research, 44(2), 147-181.

Thank you for this. While recognizing the small sample size, we also found this paper helpful and had cited it in the original manuscript (under the Lexical representational structure sub-header on pg. 20, now reference 34).

However, I think that given the small sample sizes, any differences in factor structure might be uninterpretable. If the authors want to show possible group differences as an exploratory or descriptive analysis, I again suggest that showing two correlograms, one per group, would be more likely to give the reader a gist of what might be going on.

We apologize. We weren’t quite sure what you were imagining here and are less familiar with correlograms, per se. Since the largest difference in the structure between the two groups was the loading of the phonemic decoding measure, we included a scatterplot of how that correlates with Word ID in both groups. This is now Figure 3. We hope this will highlight the gist of the take-home message here in a clearer manner.

Reviewer #2: This paper presents the results of a study that examined the existence of different reading styles in English. The majority of studies have been done on reading in English, and the consensual model, the dual route model, holds that written words can be identified via the lexical, or holistic/orthographic route, or a sub-lexical, phonological decoding route. A number of cross language/writing system studies have shown that the orthographic transparency and morphological structure of specific writing systems result in more efficient processing weighting the lexical or the sub-lexical route. In English, it has been shown that both characteristics of the words and of the reader (e.g., skill level) affect the relative use of the lexical or sublexical routes. I have several major concerns:

1. I am not exactly sure of the goal of this paper: participants were divided into a ‘holistic’ or ‘sub-lexical’ category by the degree to which reading words upside-down differed from reading words in the canonical orientation. Then, after a very extensive sifting procedure, these groups were compared on reading words forwards and backwards. I understand that the point was to see if performance on this task, using words for which the holistic and sub-lexical aspects are documented, would go in the same direction. Here lies the problem – the fact that the groups differed on the backwards reading task is trivial, as they were chosen to differ on an upside-down reading task. 

We actually agree here. We reported the results that show group differences in the backwards reading task for the sake of completeness of the model. 

The fact that they only differ on performance on the upside-down words is problematic, as it suggests that in normal word identification tasks, they do not differ. Thus, when the task is made more difficult in a specific manner, then the groups differ in how well they can compensate for the spatial distortion. But that is how the groups were created to begin with – there is no theoretical reason to think that the two spatial manipulations (upside down and backwards) are inherently different.

Here we respectfully disagree. It is true that the groups only seem to differ when the word presentation is disrupted, but that was actually hypothesized. We interpret these data patterns as suggesting that while both groups are skilled readers, they are actually using different underlying procedures for word identification. Since both groups’ procedures are efficient for them, we wouldn’t necessarily expect differences in a typical word presentation. We predicted that differences would likely emerge when we made the task more difficult, as you stated. However, it isn’t the fact that one group is worse or slower when the words were distorted that we hoped to focus on, because as you stated that was part of the group definitions. We hoped to highlight the fact that different psycholinguistic factors were better predictors of reading speed in the distorted trials. Indeed, we found that high sensitivity readers were significantly more affected by length (and marginally by imageability), both considered lexical factors, when words were distorted, but low sensitivity readers were marginally more affected by biphone frequency, considered to be a sublexical factor. While some of the results are weaker, we hoped to highlight that all observed effects are in the predicted direction, as you state below. 

To your other point regarding there being no theoretical reason that the two manipulations are inherently different, we actually agree. We wanted to use two different manipulations to strengthen the argument that different manipulations should similarly disrupt reading ability and lessen a potential concern that these effects are specific to one manipulation. We highlight this point on pg. 10.

2. The finding that the factor structure of the RTs is different in a predictable manner could be even more supporting evidence that this division is reasonable, but the very small number of participants in the groups (11 and 14) really makes it hard to believe it…-- and, it is also true by the way that the groups were defined.

We hope the response to the above comment will address some of these concerns. 

Lastly, while indeed some effects are quite small, but in the predicted directions, the 3-way (group x presentation orientation x length) interaction is highly significant (p=.000102), which is very unlikely to be due to chance, and added a reference to support that point on pg. 24 (Ioannidis JP. The proposal to lower P value thresholds to. 005. Jama. 2018;319(14):1429-30). We do not believe this effect is directly linked with how the groups were defined, as our criteria for inversion sensitivity did not include length as a factor. But we do agree with your concern about the strength of the other effects. We have tried to be more transparent in the text regarding this issue, both in how we describe the effects, and on page 23 of the discussion, where we explicitly note that some of our results are marginal. 

3. There seem to be some missed opportunities here. Although it is not mentioned, the division of word characteristics into lexical and sub-lexical categories follows results shown by many studies of reading by split-brain patients, reading by unilateral brain damaged patients, and healthy participants identifying words in divided visual field paradigms. This has been done across languages (e.g., Rao & Vaid, 2017; Zhou et al, 2019; Ibrahim & Eviatar, 2012; and many more). Given the very large initial sample, it would have been interesting to see if left-handed participants tended to fall into one or the other group. That would have been a novel result.

We did not include left-handed subjects in our sample, and so cannot pursue this suggestion. However, we appreciate this thoughtful comment and would be interesting in pursuing these ideas in future work.

Summary: Other than the very small sample size, the study seems well done. I am just not sure what it adds to the literature.

Reviewer #3: The investigators have concluded that their results revealed that greater orientation sensitivity was associated with a reading profile that relies less on sublexical phonological measures and more on lexical-level characteristics within the skilled English readers. This is based on a statistical approach involving typical procedures such as the t-test, mixed models, the empirical logit model and principal component analysis. There are some concerns. Specifically:

1. The authors note that participants were recruited from a database of 411 individuals interested in study opportunities. It is not clear what ‘study opportunities’ means here or if it is even relevant. 

We apologize for any unwanted ambiguity. The Perfetti Lab recruits several hundred participants each year for a behavioral screening in order to drive several more specific research studies where participants are needed with specific behavioral characteristics. We were conducting one of those many studies. 

They explain that a number of subjects were removed leaving 203 from an initial 411 potential participants. This appears to be a convenience sample.

We first filtered based on overall reading ability, in the hopes to avoid testing participants at the extremes of the distribution. We’re not sure this would necessarily be considered convenience sampling, which to our understanding would be recruiting only participants that were available or nearby.

A hypothesis is stated on page 7 in the form that native English readers who show more sensitivity to atypical orientation do so because they make greater use of holistic orthographic coding, which should be reflected in a distinctive reading profile. This appears to be a comparison of the low to high sensitivity group which they define. The problem is that there is no statistical design motivation for the sample size. Is 203 individuals statistically sufficient to test their hypothesis with reasonable statistical power? 

Also on the bottom of Page 8 the investigators note that participant attrition, due to graduation since participation in initial screening and eligibility requirements for a parallel imaging study (beyond the scope of the current study), also reduced the potential participant pool. What exactly does this mean and what was the final number of participants for each of the Tables 1 to 5?

That is correct. There were several extenuating circumstances that lead to a smaller sample including that we wanted to first recruit participants with nonoverlapping inversion sensitivity scores (described and highlighted on page 9), and that they also be eligible for a companion imaging study. Along with attrition from the original database due to graduation or simply not responding, the pool that we drew from was significantly smaller. We believe future research will be important to understand how common these behavioral profiles are.

The final number of participants for all tables was 14 for LS readers and 11 for HS readers.

2. As minor points please define RT on page 8. It appears to be 'reaction time'. On the top line of page 20 the term ‘ phonological decoding (.728)’ should be ‘Phonemic Decoding (0.728)’ as per Table 5.

Thank you- these corrections have been made.

---

## [Decision Letter · Decision Letter 1]

24 Jan 2020

PONE-D-19-17391R1

Unmasking individual differences in adult reading procedures by disrupting holistic orthographic perception

PLOS ONE

Dear Dr. Hirshorn,

Thank you for submitting your manuscript to PLOS ONE. After careful consideration, we feel that it has merit but does not fully meet PLOS ONE’s publication criteria as it currently stands. Therefore, we invite you to submit a revised version of the manuscript that addresses the points raised during the review process.

I sent your revised manuscript to one of the original reviewers and to one new reviewer as the other reviewers were no longer available. As you will see below both reviewers commented that the manuscript has been improved and  suggested minor changes to the current version. I therefore invite you to address these comments before we will able able to move forward and accept your paper for publication. 

We would appreciate receiving your revised manuscript by Mar 09 2020 11:59PM. To enhance the reproducibility of your results, we recommend that if applicable you deposit your laboratory protocols in protocols.io, where a protocol can be assigned its own identifier (DOI) such that it can be cited independently in the future. For instructions see: http://journals.plos.org/plosone/s/submission-guidelines#loc-laboratory-protocols

We look forward to receiving your revised manuscript.

Kind regards,

Yafit Gabay

Academic Editor

PLOS ONE

Reviewers' comments:

Reviewer's Responses to Questions

**Comments to the Author**

1. If the authors have adequately addressed your comments raised in a previous round of review and you feel that this manuscript is now acceptable for publication, you may indicate that here to bypass the “Comments to the Author” section, enter your conflict of interest statement in the “Confidential to Editor” section, and submit your "Accept" recommendation.

Reviewer #1: (No Response)

Reviewer #4: (No Response)

2. Is the manuscript technically sound, and do the data support the conclusions?

Reviewer #1: Partly

Reviewer #4: Yes

3. Has the statistical analysis been performed appropriately and rigorously? 

Reviewer #1: No

Reviewer #4: Yes

4. Have the authors made all data underlying the findings in their manuscript fully available?

Reviewer #1: Yes

Reviewer #4: Yes

5. Is the manuscript presented in an intelligible fashion and written in standard English?

Reviewer #1: Yes

Reviewer #4: Yes

6. Review Comments to the Author

Reviewer #1: Thank you, my comments are below:

Regarding the comment of reviewer #2: „...the fact that the groups differed on the backwards reading task is trivial, as they were chosen to differ on an upside-down reading task.“ I started looking into this, and what I missed in my first review of the paper is that the authors apparently not only assessed the orientation sensitivity for the second time, but they also redefined their groups based on this dependent measure: „Six participants whose overt naming scores were neither above nor below the median orientation sensitivity that was consistent with their initial group assignment based on the lexical decision task were removed.“ I don‘t think that this is a good idea, if I understand correctly what the authors did, as this can be considered double dipping into the data. As this was done, I don‘t think that the main effects of orientation or interactions with orientation should be trusted, although I suggest getting a second opinion from a statistician. I suggest rerunning all models with these participants included. Otherwise, group assignments and orientation effects are not independently assessed.

What is going on in the new figure 2 in terms of RTs of atypical word presentations? Almost all of the RTs for the low sensitivity (red) seem to fall under the red regression line while almost all of the RTs for the high sensitivity (cyan) seem to fall under the cyan regression line. I suspect that this has something to do with the actual drawing of the graph, i.e. perhaps almost all of the red dots on the top are underneath the cyan dots on the top. This could likely be amended by making them partially transparent.

Also, the x-axis of figure 2 just says value.z. I suggest changing this to something a bit more transparent, and saying in the figure legend that the x-axis represents each psycholinguistic factor.

Finally, for that graph, please say explicitly that these are (if they are) RTs from individual trials pooled across participants.

The statistics in Table 1 still look weird. The t-value(s) and p-value(s) still don‘t match. I am assuming an N of 14 LS and 11 HS.

Reviewer #4: The authors show that greater sensitivity to orientation/greater holistic processing is associated more with lexical than sub-lexical characteristics, thus holistic strategies seem to be involved with lexical reading strategies.

I was not one of the three original reviewers. My fellow colleagues did an excellent job in their revisions and I also consider the response of the authors adequate. I have only a few points that the authors might additional take into consideration.

(i) One question is the very small number of participants. I wonder if the authors could provide a power analysis study using e.g., g*power.

(ii) An indicator of holistic word processing—observers’ sensitivity to changes in configural/spatial jittering information of objects in an inversion paradigm has been used by Wong et al. (2019). Is this manipulation closer to what is meant by holistic processing? Just inverting the stimuli maybe not be enough to study holistic processing.

(iii) Sensitivity to atypical orientation in the overt naming task used half of the stimuli presented in typical orientation and half presented in a reversed (FLIGHT � THGILF) orientation. What has this to do with holistic strategy

(iv) Authors hypothesize that the HS group should be more affected than the LS group by lexical-level factors when words are atypically oriented, whereas LS should be more affected than HS by sublexical-level factors when words are atypically oriented. But is it equally possible to make the same type of predictions for when words are typically oriented and lexical factors more at play?

7. PLOS authors have the option to publish the peer review history of their article (what does this mean?). If published, this will include your full peer review and any attached files.

Reviewer #1: No

Reviewer #4: No

---

## [Author Response · Author response to Decision Letter 1]

14 Apr 2020

This information is also in the Response to Reviewers document that has been uploaded with better visuals and formatting:

Dear Dr. Yafit Gabay and Reviewers,

First, we apologize for the delay in response as we adjusted to impending and current changes in daily life. We very much appreciate the opportunity to address the questions from past and new reviewers and believe they have greatly improved the paper again. Responses to individual comments are below in blue italics. Many thanks to all, and we hope everyone is safe and healthy.

Sincerely, 

Elizabeth Hirshorn, PhD

Reviewer #1:

1. Regarding the comment of reviewer #2: „...the fact that the groups differed on the backwards reading task is trivial, as they were chosen to differ on an upside-down reading task.“ I started looking into this, and what I missed in my first review of the paper is that the authors apparently not only assessed the orientation sensitivity for the second time, but they also redefined their groups based on this dependent measure: „Six participants whose overt naming scores were neither above nor below the median orientation sensitivity that was consistent with their initial group assignment based on the lexical decision task were removed.“ I don‘t think that this is a good idea, if I understand correctly what the authors did, as this can be considered double dipping into the data. As this was done, I don‘t think that the main effects of orientation or interactions with orientation should be trusted, although I suggest getting a second opinion from a statistician. I suggest rerunning all models with these participants included. Otherwise, group assignments and orientation effects are not independently assessed.

Thank you for this comment and the chance to consider this concern. We understand the concern of ‘double dipping,’ and agree that the 2-way interaction between group and orientation is a trivial finding, since that is essentially how the groups were defined. We consulted with a statistician, who confirmed that ‘double-dipping’ should not affect the direction or strength of the 3-way interactions, since there is not anything inherent about being sensitive to inversion, per se, that would cause one group to be more influenced than another by a psycholinguistic measure when words were presented atypically. In fact, that is what we were interested in testing. Furthermore, words that were presented in the typical and atypical orientations were matched for all psycholinguistic factors (e.g., there weren’t more long words that were atypical than typical). 

Nevertheless, we did take your advice and added in the previously removed participants and ran the model again with “group” as a continuous variable based solely on the lexical decision task (LDT), which eliminates the double-dipping component of group selection since it doesn’t incorporate the naming task at all. The 3-way interaction between “group” (i.e., strength of inversion sensitivity in LDT task) x presentation orientation x length was actually even more significant with this approach (see Table below- yellow highlighted row). 

Specifically, the previous t-value for that 3-way interaction was 3.89, and adding the other participants led to a t-value of 6.12. We believe this provides further support that the group/sensitivity x orientation x length 3-way interaction is robust. The two marginal 3-way interactions did not hold up (see response about power for Reviewer #4 below), but another marginal 3-way interaction emerged in the predicted direction, such that bigram frequency was a marginally better predictor of RT for atypically presented words in those whose LDT were relatively small (i.e., those with low sensitivity). We previously discussed our marginal effects in a tentative manner, so this change in marginal results does not undermine our results. Instead, we believe that the fact that a conceptually related marginal effect emerges supports the same weak overall pattern of psycholinguistic differences between the two groups.

Despite the strengthening of the group x orientation x length effect in the updated model, we are proposing to keep the main reporting of results the same, as a group design, but propose to add the results of the additional analysis for transparency (see pg. 17-18). We propose to keep the main focus on the original group-based analysis for the following reasons:

• Our a-priori hypotheses and design were meant for having separate groups.

• Related to reviewer #4’s concern that inverting words alone may not be enough to measure ‘holistic processing,’ we believe that using two similar, but not identical measures actually strengthens different aspects of the study.

o For example, based on logistics and timing, the LDT experiment did not have a lot of trials and we wanted to reduce any potential noise by combining it with an additional task that was different, but conceptually similar in order to have more stability in our measurement. 

o We recently published a paper (Carlos et al, 2019) that used this same grouping. In this paper the same Ss were used to look at group differences in a neural measure of lateralization, with more inversion sensitive participants exhibiting a more bilateral pattern of activation in a putative visual word form area. This result provides converging evidence that our groups exhibit different patterns of orthographic processing, with no concerns about “double-dipping” into the same data.

o Since these two papers are in essence companion pieces, we believe it is more straightforward to be able to talk about the same participants.

That being said, our current and future plans for the continuation of this research will be using more continuous measures of sensitivity. 

What is going on in the new figure 2 in terms of RTs of atypical word presentations? Almost all of the RTs for the low sensitivity (red) seem to fall under the red regression line while almost all of the RTs for the high sensitivity (cyan) seem to fall under the cyan regression line. I suspect that this has something to do with the actual drawing of the graph, i.e. perhaps almost all of the red dots on the top are underneath the cyan dots on the top. This could likely be amended by making them partially transparent.

Thank you for sharing this observation. The pattern that you noticed could be partially a consequence of the group difference in RT for atypically presented words (low sensitivity readers are faster than high sensitivity readers by design), but we understand what you’re saying. At your suggestion, we did make a version with more transparent markers, but unfortunately it just ended up looking extremely blurry and harder to interpret. For the sake of perceptual clarity, we think the information is better represented with more opaque markers, albeit less than ideal. We believe that the main take-away message from this figure, to see the non-parallel lines in the atypical presentation for length (plus imageability & biphone frequency) should still be able to be communicated.

2. Also, the x-axis of figure 2 just says value.z. I suggest changing this to something a bit more transparent, and saying in the figure legend that the x-axis represents each psycholinguistic factor.

Thank you for this suggestion. This was changed, so that the x-axis is now titled, ‘z-score of Each Psycholinguistic Factor,’ thank you.

3. Finally, for that graph, please say explicitly that these are (if they are) RTs from individual trials pooled across participants.

Thank you for this suggestion. This information was added to the figure caption.

4. The statistics in Table 1 still look weird. The t-value(s) and p-value(s) still don‘t match. I am assuming an N of 14 LS and 11 HS.

Thank you for pointing this out. We regret that the previous version did indeed have errors in Table 1. The small edits needed to correct these errors have been made. Importantly, the results are essentially the same.

Reviewer #4

The authors show that greater sensitivity to orientation/greater holistic processing is associated more with lexical than sub-lexical characteristics, thus holistic strategies seem to be involved with lexical reading strategies. I was not one of the three original reviewers. My fellow colleagues did an excellent job in their revisions and I also consider the response of the authors adequate. I have only a few points that the authors might additional take into consideration.

1. One question is the very small number of participants. I wonder if the authors could provide a power analysis study using e.g., g*power.

We understand the concern about a small sample. We consulted with a statistician, and instead of using g*power, we used a function in R that is suited to linear mixed models (see powerSim in simr package). This allows us to estimate the power for specific effects in our model using Monte Carlo estimation:

Green, P., & MacLeod, C. J. (2016). SIMR: an R package for power analysis of generalized linear mixed models by simulation. Methods in Ecology and Evolution, 7(4), 493-498.

Brysbaert, M., & Stevens, M. (2018). Power analysis and effect size in mixed effects models: A tutorial. Journal of Cognition, 1(1).

We ran this for each of the three 3-way interactions that showed effects, set at 200 simulations. The 3-way interaction between group x orientation x length had a very large effect size, and the observed power was 96.50% (95% confidence interval: 92.92, 98.58). Not surprisingly, the two marginal effects had much lower power- both had 51% power (43.85, 58.12). We hope that talking about these effects as more tentative will be adequate.

We have modified the manuscript to include this power analysis in the results section (pg. 17), and have retained our previous caution in presenting and discussing our marginal effects.

2. An indicator of holistic word processing—observers’ sensitivity to changes in configural/spatial jittering information of objects in an inversion paradigm has been used by Wong et al. (2019). Is this manipulation closer to what is meant by holistic processing? Just inverting the stimuli maybe not be enough to study holistic processing.

Thank you for this comment, as this is an ongoing issue in the field. In fact, since this work has been partially inspired by the face processing literature, we share the concern that there is not necessarily one manipulation that completely encompasses the construct of ‘holistic’ processing (see Rezlescu et al, 2017). That is one of the main reasons we chose to combine slightly different measures for group assignment.

Rezlescu, C., Susilo, T., Wilmer, J. B., & Caramazza, A. (2017). The inversion, part-whole, and composite effects reflect distinct perceptual mechanisms with varied relationships to face recognition. Journal of Experimental Psychology: Human Perception and Performance, 43(12), 1961.

That being said, the current and future work in this research program uses a combination of measures, including a composite task. For this paper, we have added a paragraph to our discussion section (pg. 28) that notes this important issue, with citations to Wong et al. (2109) and Rezlescu et al. (2017) included.

3. Sensitivity to atypical orientation in the overt naming task used half of the stimuli presented in typical orientation and half presented in a reversed (FLIGHT à THGILF) orientation. What has this to do with holistic strategy?

Our thinking is that any perceptual manipulation that disrupts the typical processing of an object should also disrupt holistic processing, if that is used. For example, other studies have used case, size, and font alteration to add visual noise and thus disrupt ‘holistic’ processing (see Pae et al, 2017). 

Pae, H. K., Kim, S. A., Mano, Q. R., & Kwon, Y. J. (2017). Sublexical and lexical processing of the English orthography among native speakers of Chinese and Korean. Reading and Writing, 30(1), 1-24.

While our reverse presentation is not a widely used manipulation, we consider it to be a conceptual extension of previous manipulations (such as case alternation) – that is, the idea that presenting a word in any atypical orientation should disproportionately disrupt those who rely more on a ‘holistic’ reading procedure.

We note that similar ideas are present in the face and object processing literature. For instance, Perret, Oram, and Ashbridge (1998) propose that the speed of object recognition rests upon the accumulation of activity from neurons selective for the object as experienced in a particular viewing circumstance. This neuronal tuning is presumed to be sensitive to frequency of occurrence, and so the neural activity builds more quickly for objects presented in a canonical as compared to unusual view.

4. Authors hypothesize that the HS group should be more affected than the LS group by lexical-level factors when words are atypically oriented, whereas LS should be more affected than HS by sublexical-level factors when words are atypically oriented. But is it equally possible to make the same type of predictions for when words are typically oriented and lexical factors more at play?

This is a very astute point. The reason we focused on atypically oriented words is that the effects may be smaller and not as easily measured when words are typically oriented, since both groups are skilled and efficient using whatever reading procedures are most natural. In contrast, with an atypical orientation the idea is that all participants are forced to rely more upon sublexical procedures, and so this helps to “unmask” differences in the ability to use these procedures.

However, we do agree that biases for the different procedures remain present when words are presented in their typical orientation. In support of this point, in a previous paper, we compared Chinese- and Korean-English bilinguals: 

Ben-Yehudah, G., Hirshorn, E. A., Simcox, T., Perfetti, C. A., & Fiez, J. A. (2019). Chinese-English bilinguals transfer L1 lexical reading procedures and holistic orthographic coding to L2 English. Journal of Neurolinguistics, 50, 136-148.

As we discuss in the introductory section of our manuscript, in this prior work we tested our prediction that Chinese-English bilinguals would use a more holistic reading procedure than Korean-English bilinguals. In support of this hypothesis, we found that the Chinese-English bilinguals exhibited greater sensitivity to inversion than the Korean-English bilinguals. 

One finding from this paper that we did not previously discuss concerned differences in the reading skill profile of the two bilingual groups. Even though we matched the groups on the basis of spoken English experience and ability, we found that the Chinese-English bilinguals exhibited performance differences on typically presented items. For instance, they performed more poorly on the Word Attack and Word ID subtests of the Woodcock Reading Mastery Tests, and we found that the Chinese-English bilinguals exhibited greater sensitivity to lexical frequency on a word-naming task that included upright and inverted items. In the discussion section of Ben-Yehudah et al. paper, we discuss the implication of these results, which are similar to those reported in previous studies. Specifically, we suggest that Chinese-English bilinguals have a bias towards a holistic procedure is a less optimal approach for developing reading skill in an alphabetic writing system. Consequently, they require more reading experience to attain the same level of skill.

These same ideas could be relevant for native English speakers to exhibit a bias towards a holistic procedure. However, we would be unlikely to find them in the present study, because we selected the two groups to be matched on measures of reading skill. If we instead could have selected the group to be matched on reading experience or randomly selected them without concern for reading skill, then it is very possible that we would have observed the pattern suggested by the reviewer. 

Due to concerns about the overall length of the paper, we have not revised our discussion section to consider this point. However, if the reviewer feels that this is an issue that merits consideration, we would be pleased to add text addressing the reviewer’s comment.

---

## [Decision Letter · Decision Letter 2]

28 Apr 2020

Unmasking individual differences in adult reading procedures by disrupting holistic orthographic perception

PONE-D-19-17391R2

Dear Dr. Hirshorn,

We are pleased to inform you that your manuscript has been judged scientifically suitable for publication and will be formally accepted for publication once it complies with all outstanding technical requirements.

With kind regards,

Yafit Gabay

Academic Editor

PLOS ONE

Additional Editor Comments (optional):

Reviewers' comments:

Reviewer's Responses to Questions

**Comments to the Author**

1. If the authors have adequately addressed your comments raised in a previous round of review and you feel that this manuscript is now acceptable for publication, you may indicate that here to bypass the “Comments to the Author” section, enter your conflict of interest statement in the “Confidential to Editor” section, and submit your "Accept" recommendation.

Reviewer #1: All comments have been addressed

2. Is the manuscript technically sound, and do the data support the conclusions?

Reviewer #1: Yes

3. Has the statistical analysis been performed appropriately and rigorously? 

Reviewer #1: I Don't Know

4. Have the authors made all data underlying the findings in their manuscript fully available?

Reviewer #1: Yes

5. Is the manuscript presented in an intelligible fashion and written in standard English?

Reviewer #1: Yes

6. Review Comments to the Author

Reviewer #1: (No Response)

7. PLOS authors have the option to publish the peer review history of their article (what does this mean?). If published, this will include your full peer review and any attached files.

Reviewer #1: No

---

## [Editor Report · Acceptance letter]

4 May 2020

PONE-D-19-17391R2 

Unmasking individual differences in adult reading procedures by disrupting holistic orthographic perception 

Dear Dr. Hirshorn:

I am pleased to inform you that your manuscript has been deemed suitable for publication in PLOS ONE. Congratulations! Your manuscript is now with our production department. 

With kind regards,

on behalf of

Dr. Yafit Gabay 

Academic Editor

PLOS ONE